# The Specific Gravity-Free Method for the Isolation of Circulating Tumor KRAS Mutant DNA and Exosome in Colorectal Cancer

**DOI:** 10.3390/mi12080987

**Published:** 2021-08-20

**Authors:** Tae Hee Lee, Eunsook Park, Young-gon Goh, Han Byul Lee, Woo Sun Rou, Hyuk Soo Eun

**Affiliations:** 1Research Institute for Future Medical Science, Chungnam National University Sejong Hospital (CNUSH), Sejong 30099, Korea; youngon@cnuh.co.kr (Y.-g.G.); lhbskans@naver.com (H.B.L.); 2Department of Chemistry, Korea Advanced Institute of Science and Technology (KAIST), Daehak-ro, 291, Daejeon 34141, Korea; sjil6273@kaist.ac.kr; 3Division of Gastroenterology and Hepatology, Department of Internal Medicine, Chungnam National University Sejong Hospital, Sejong 30099, Korea; rws00@cnuh.co.kr; 4Division of Gastroenterology and Hepatology, Department of Internal Medicine, Chungnam National University Hospital, 282, Munwha-ro, Jung-gu, Daejeon 35015, Korea; 5Department of Internal Medicine, College of Medicine, Chungnam National University, 266, Mun-wha-ro, Jung-gu, Daejeon 35015, Korea

**Keywords:** specific gravity, circulating tumor DNA (ctDNA), KRAS, exosome

## Abstract

Background: Circulating tumor DNA (ctDNA) and exosome have been widely researched in the field of medical technology and diagnosis platforms. The purpose of our study was to improve the capturing properties of ctDNA and exosome, which involved combining two beads using approaches that may provide a new method for cancer diagnoses. Methods: We present a dual isolation system including a polydopamine (PDA)–silica-coated alginate bead for circulating tumor DNA (ctDNA) capture and an anti-CD63 immobilized bead for exosome capture. We examined the ctDNA mutation in pre-operative plasma samples obtained from 91 colorectal cancer (CRC) patients using a droplet digital PCR (ddPCR). Results: The area under the curve (AUROC) of ctKRAS G12D mutation in the buffy coat was 0.718 (95% CI: 0.598−0.838; *p* = 0.001). Patients with CRC that had unmethylation of MLH1 and MSH2 showed significantly higher buffy coat ctKRAS G12D mutations, ascites ctKRAS G12D mutations, miR-31-5, and mixed scores than the patients with a methylation of MLH1 and MSH2. Conclusion: Our proposed alginate bead using the specific gravity-free method suggests that the screening of mutated ctKRAS DNA and miR-31-5 by liquid biopsy aids in identifying the patients, predicting a primary tumor, and monitoring in the early detection of a tumor.

## 1. Introduction

Colorectal cancer (CRC) is the second most common cancer leading to death in the US. A colonoscopy has several disadvantages, which include intestinal cleanliness needed to perform the procedure along with pain endured for surveillance, and, in particular, colorectal cancer patients need a regular colonoscopy evaluation for recurrence, so the procedure might have to frequently be performed [1]. Since this test is relatively invasive, liquid biopsy can be a good alternative diagnostic method. Tumor markers, such as carbohydrate antigen 19-9 (CA19-9) or carcinoembryonic antigen (CEA), have widely been used for predicting a prognosis [2]. However, CA19-9 and CEA are not useful for predicting a prognosis immediately after surgery. The identification of new biomarkers reflecting the current disease activity is urgently needed for CRC. Indeed, a KRAS mutation observed from a primary tumor is one of the most widely studied chromosomal instabilities for CRC. Compared with a KRAS mutation from a primary tumor, ctDNA and exosome is a less invasive method of analyzing the genomic profiles of mutations across the tumor gene for monitoring from the patients [3,4]. Furthermore, genetic information in the tumor of an individual patient can be rapidly characterized by analyzing the ctDNA and exosome, which are released into the blood circulation through the apoptosis or necrosis of cancer cells [5]. Besides blood, ctDNA and exosome are detected in various body fluids, including ascites, urine, and saliva [6,7]. Here, we suggest that miR-31-5 in the exosome and a ctKRAS G12D mutation from ascites and the buffy coat may provide additional information, and these were detected by tumor and molecular characterization [8,9,10]. Furthermore, microRNA (miRNA) expression in the exosome are investigated between KRAS-mutated CRCs and MLH1/MSH2-methylated CRCs, since miRNAs were found to be upregulated or downregulated in CRC tissues or serum compared to healthy tissue [8,9,11]. To capture the ctDNA and exosome, we first chose an alginate polymer as a particle source due to its unique curing property where it was ionically crosslinked with a multivalent metal ion (e.g., calcium ion and iron ion), resulting in the formation of a hydrogel [12,13]. To provide a silica surface for DNA absorption, we firstly adopted a polydopamine coating. The PDA efficiently interconnected the polymeric alginate backbone and silica particles with its intrinsic material-independent adhesive property [14]. In our previous study, we demonstrated the superior DNA absorption capability using our PDA–silica-coated alginate bead [15]. Secondly, exosome isolation was generally required for the large sample volume quantities and high-specific-gravity, such as ultracentrifugation at 100,000× *g* [16]. Since it contains drawbacks, the proposed alginate bead does not require high-specific-gravity and covers large sample volume quantities efficiently. We used an anti-CD63-mediated bead for the specific exosome capture rather than the large volume or high-gravity system.

In the present study, we combined these two strategies for efficient mutation detection and as a monitoring tool for CRC patients, and claimed the usefulness of the utility of liquid biopsy in the prediction of a primary tumor. The relationship among ctKRAS G12D mutation in the buffy coat, ctKRAS G12D mutation in ascites, miRNA-31-5, and KRAS immunohistochemistry (IHC) results in the tissues of origin were assessed to demonstrate that ctDNA isolated by a PDA–silica-coated alginate bead and by anti-CD63 immobilized alginate beads could reflect molecular characteristics of the primary tumor.

## 2. Materials and Methods

### 2.1. Preparation of the Coated Alginate Bead of PDA–Silica and Anti-CD63

Sodium alginate (5% *w*/*v*) was dissolved in distilled water (DW) and then dropped into 100 mM of calcium chloride aqueous solution for bead formation via Ca^2+^-mediated curing (1 h, room temperature (RT)). The alginate beads were carefully washed three times with DW and then stored in DW until further use. The one bead group was prepared by PDA coating, followed by silica coating. In brief, alginate beads were firstly coated in 5 mM EDC/NHS for 1 h at room temperature and then washed ten times with DW. Then, the beads were subsequently coated with the PDA by incubation in the dopamine hydrochloride solution (5 mM in Tris-HCl buffer, pH ~6.5) for 12 h at room temperature. PDA–silica-coated alginate beads were rinsed with DW three times and then mixed with 1 µL silica solution for 1 h at RT. The resulting PDA–silica-coated alginate beads were immediately used for ctDNA capture. The other alginate bead group was coated with anti-CD63. First, the surface of the alginate beads was modified with EDC/NHS chemistry. Carboxylic groups on the alginate bead were activated with 5 mM of EDC/NHS for 1 h at RT. 

After being rinsed with PBS buffer solution, the anti-CD63 antibody (1:10, abcam; TS63, Cambridge, UK) was incubated at 4 °C for 12 h. The unreacted antibody was washed with PBS buffer solution. The anti-CD63-coated alginate beads were immediately used for the exosome isolation.

### 2.2. Surface Characterization Using SEM/EDS

In order to confirm the morphology and coated Si, the morphology of the presented beads was examined by scanning electron microscopy with energy-dispersive spectrometry (SEM/EDS) using an SU8200 (Hitachi, Japan). The prepared bead samples were attached to double-sided adhesive tape, mounted on a SEM stub and were coated with osmium of 3.0 nm thickness to avoid charging effects. Surface morphology was obtained at an acceleration voltage of 2 kV with a working distance of 4.7 mm. All images were magnified by a factor of 1.00 K. In addition, elemental composition and distribution was analyzed using the Octane Elite Super EDS System (AMETEK, Inc., Berwyn, PA, USA), which incorporates a silicon nitride (Si_3_N_4_) window. The EDS mapping was achieved at 10 kV, with a 124.5 eV resolution, over 30 s. The take-off angle of the photoelectron was set at 29.1°. Each measurement was repeated for the selected area and the results were analyzed by TEAM™ EDS Analysis System (AMETEK, Inc., Berwyn, PA, USA).

### 2.3. Isolation of Circulating Tumor DNA (ctDNA) and Exosome 

The patient samples were provided by the National University Hospital Biobank of Chungbuk, a member of the Korea Biobank Network. This study was approved by an institutional review board (IRB) of Chungnam National University Sejong Hospital, Sejong, Korea (CNUSH-20-11-012). The pre-treatment process for DNA extraction of the sample is as follows: 200 µL of buffy coat, following the removal of erythrocyte and ascites, was treated with 20 µL of proteinase K. The reaction ended and the buffy coat and ascites were mixed with 200 µL of lysis buffer by pulse-vortexing for 15 s, then incubated at 57 °C for 15 min. The sample was added to 200 µL of 95% ethyl-alcohol (Samchun, Seoul, Korea) and mixed again by pulse-vortexing for 15 s. The bead was reacted with the pretreated sample for 10 min. The bead was carefully dipped in 350 µL of buffer AW1 (Qiagen) for 60 s, then dipped in 50 µL of RNase/DNase-free water for 3 min. The elution was stored at −80 °C. For microRNA (miRNA) extraction, the isolated exosome using alginate beads was added to a 0.5 M EDTA solution (Invitrogen, Carlsbad, CA, USA). After release to the alginate bead, miRNA was extracted using the TRIzol method. A volume of 500 µL of TRIzol reagents were added to the samples from exosomes. The samples were mixed by pipetting for 3 min, then incubated for 5 min at room temperature. Using a shaker, 100 µL of chloroform was mixed thoroughly for 15 s, followed by 10 min of incubation at room temperature. Phase separation was then conducted on centrifugation at 13,000 rpm at 4 °C for 10 min. The superior aqueous phase was mixed with 500 µL of isopropanol (IPA). The samples were vortexed and incubated at RT for 10 min. The samples were centrifuged once more at 13,000 rpm at 4 °C for 10 min. The supernatants were aspirated. To remove phenol, 600 µL of 70% ethanol was added to the sample. Ethyl alcohol was then removed by centrifugation at 13,000 rpm at 4 °C for 5 min. The sample was further processed by air-drying the exosome NA pellet for 5 min. The pellets were re-suspended to 50 µL of RNase/DNase free DW and stored at −70 °C, before further analysis.

### 2.4. Droplet Digital PCR Workflow

The ddPCR technique allows the absolute quantification and rare allele detection by partitioning individual DNA copies into microdroplets of oil emulsion. Droplet digital polymerase chain reaction (ddPCR; QX200, Bio-Rad, Hercules, CA, USA) was used in this study, and each ddPCR assay mixture was performed in 20 µL of reaction volume. The mixture consisted of up to 30 ng of extracted DNA (1 µL), 2X EvaGreen ddPCR Supermix (10 µL), KRAS G12D forward: 5′-GGTGGAGTATTTGATAGTGTATTAACC-3′, reverse: 5′-AGAATGGTCCTGCACCAGTAA-3′ individually (1 µL), and DW (7 µL). The ddPCR assay mixture was loaded into a disposable droplet generator cartridge. Next, 70 µL of droplet generation oil was loaded into each of the eight oil wells. The cartridge was then placed inside the QX200 droplet generator. When droplet generation was completed, the droplets were transferred to a 96-well PCR plate. The plate was heat-sealed with foil and placed in a conventional thermal cycler (bio-Rad, T100), and was cycled with the following conditions: 95 °C for 5 min (1 cycle); 95 °C for 30 s and 55 °C for 1 min (40 cycles); 4 °C for 5 min, 90 °C for 5 min (1 cycle), 4 °C hold. Cycled droplets were read individually with the QX200 droplet-reader. A no-template control (NTC) and a negative control for each reverse transcription reaction were included in every assay.

### 2.5. Statistics

The clinical data of biomarkers, including KRAS, BRAF, MLH1, MSH2, NRAS, EGFR, CEA, and CA19-9, were compared based on Student’s *t* test or Mann−Whitney *U* test, depending on the normality of a sample distribution. The mixed score was defined by the calculation: [(X _ctKRAS DNA in buffy coat_ − average _ctKRAS DNA in buffy coat_)/(Standard deviation_ctKRAS DNA in buffy coat_)] + [(X_ctKRAS DNA in ascite_ − average _ascite_)/(Standard deviation_ascite_)] + [(X_mir-31-5_ − average _mir-31-5_)/(Standard deviation_mir-31-5_)]. Clinical capabilities of these biomarkers were further assessed by a receiver operating characteristic (ROC) curve. A value of *p* < 0.05 was regarded as statistically significant. All statistical analyses were made through SPSS Statistics 26 (SPSS, Chicago, IL, USA).

## 3. Results

### 3.1. Immobilized Alginate Bead for ctDNA Absorption and Exosome Isolation

We first prepared the alginate beads by dropping an alginate solution into a beaker containing calcium ions. The alginate was immediately cured by the calcium ions to form gel-type beads. We split the alginate beads into two groups: one alginate bead group was coated with an anti-CD63 for miR31-5 isolation and the other bead group was coated with polydopamine and silica for ctDNA isolation (Figure 1). The anti-CD63-coated bead was shown in the opaque white color by calcium-mediated crosslinking; however, the PDA–silica-coated alginate bead was shown in the black color by dopamine oxidation. Moreover, to demonstrate the coating of PDA and silica of the alginate beads, the surface of the original alginate beads and the coated beads were characterized by scanning electron microscopy (SEM) and energy dispersive spectroscopy (EDS) (Figure 2). Importantly, we confirmed a rough surface of beads, due to the silica particles after sequential coating with PDA and silica (Figure 2c). Indeed, the EDS results clearly support the silica-coated surface. Compared with the original alginate beads which showed the high contents of Ca and Cl due to the initial calcium curing steps (Figure 2b), the PDA–silica-coated alginate beads exhibited high contents of Si and O (Figure 2d). The smooth coating of the Si particle on the bead might be attributed to the strong adhesion property of PDA [12,13,14]. Furthermore, these results implied that adsorped cfDNA by PDA enhanced the additional molecular gene information.

### 3.2. Clinical Characterization in CRC Patients

The presented study included 91 patients with clinical data of KRAS, BRAF, MLH1, MSH2, and NRAS expression. The clinical and pathological features representative of our patient cohort are described in Table 1. We identified 91 CRC patients between the ctKRAS G12D mutation and miR-31-5 expression. Mutation of ctKRAS G12D in the buffy coat and ascites occurred in 44/91 (≥median, 48.4%) and 44/91 (≥median, 48.4%) of cases, respectively. miR-31-5 expression in the exosome was found in 46/91 (≥median, 50.5%) of cases.

### 3.3. The Diagnostic Capability of ctKRAS G12D Mutation and miR-31-5 from CRC Patients

The application of liquid biopsy is for the detection of recurrence earlier than the currently used surveillance tests, such as: radiographic imaging; colonoscopy; and blood tumor markers such as CEA and CA19-9. The liquid biopsy for ctDNA and miRNA screening can be conducted using several approaches, including next-generation sequencing, digital PCR, and real-time PCR. Taking into consideration the cost-effectiveness in clinical practice, we applied ddPCR assays in the analysis for the first screening of pre-operative samples of CRC patients. We investigated whether the detection of mutated ctDNA and miRNA could be used as a predictive biomarker for the earlier detection of primary tumor status. Previous studies have already suggested that the detection of mutated ctDNA is an accurate prognostic biomarker [17,18]. In the present study, ctKRAS G12D mutation and miR-31-5 expression were analyzed from ddPCR (Appendix A). Figure 3 and Table 1 show the diagnostic performance of the various modalities of differential diagnosis in tissue vs. ctKRAS G12D mutation in the buffy coat, ctKRAS G12D mutation in ascites, miR-31-5 in exosome, and the mixed score. For that, ctKRAS G12D mutation in the buffy coat, ctKRAS G12D mutation in ascites, miR-31-5 in exosome, and the mixed score reflect the pathological status of the tumor burden (n = 91), and the area under the curve (AUROC) (95% CI; *p* value) was 0.718 (0.598–0.838; 0.001), 0.611 (0.489–0.734; 0.083), 0.569 (0.434–0.703; 0.286), and 0.698 (0.575–0.822; 0.002), respectively (Figure 3). For the MLH1 and MSH2 methylation in tissue vs. ctKRAS G12D mutation in the buffy coat, ctKRAS G12D mutation in ascites, miR-31-5 in exosome, and mixed score (n = 91), the area under the curve (AUROC) (95% CI; *p* value) was 0.300 (0.186–0.414; 0.003), 0.277 (0.168–0.387; 0.001), 0.293 (0.170–0.415; 0.002), and 0.269 (0.165–0.374; 0.001), respectively (Appendix A). The diagnostic accuracies of ctKRAS G12D mutation in the buffy coat, ctKRAS G12D mutation in ascites, miR-31-5 in exosome, and the mixed score were 47.1%, 69.2%, 69.1%, and 54.2% from NRAS mutation, and 54.0%, 19.3%, 55.0%, and 82.5% from BRAF mutation, respectively. However, these results were not statistically significant (*p* > 0.050). Previous work has shown that BRAF or NRAS mutant types seldom occur in the KRAS mutant type CRC patient group. In other words, in KRAS mutation in tissue, it is rarely a coincidence with other mutations such as NRAS, BRAF, and EGFR [19,20,21]. For these reasons, the KRAS status of liquid biopsy tends to be less reflective in NRAS, BRAF, EGFR, and MSI of the origin tumor.

Importantly, the study implied that the detection of ctDNA could improve the early prediction of primary tumor status by 71.8%, suggesting the necessity of periodical liquid biopsy of ctDNA screening to increase the prediction rate for primary tumor status. ROC analysis elicited a high accuracy of ctKRAS G12D mutation in the buffy coat for prediction of the KRAS mutation in tissue. Therefore, it is suggested that the ctKRAS G12D mutation in the buffy coat could be applied as a surrogate biomarker for KRAS mutation and is a better diagnostic marker for the diagnosis of liquid biopsy compared from ascites in CRC patients.

### 3.4. Liquid Biopsy of the Buffy Coat, Ascites, and Exosome Correlate with Primary Tumor Status

To investigate whether the detection of a ctKRAS G12D mutation and miR-31-5 was associated with clinicopathological parameters, we analyzed the correlation and accuracy. There was a positive correlation between the ctKRAS G12D mutation in the buffy coat and ascites of the 91 CRC patients (Pearson’s r = 0.249 and *p* = 0.017) (Appendix A). As shown in Appendix A, Pearson’s correlation coefficients of miR-31-5 from the buffy coat and ascites were 0.192 (*p* = 0.068) and 0.441 (*p* < 0.001), respectively. It also revealed that ctDNA reflected the molecular characteristics of miRNA and pathological status of the primary tumor, although Pearson’s correlation coefficient levels were low. The mutation of ctKRAS G12D DNA from the buffy coat was statistically significant from the KRAS status of the primary tumor, which was defined by IHC analysis.

The KRAS mutant in patients from a primary tumor was significantly higher in the ctKRAS G12D mutation in the buffy coat compared with those of ctKRAS wild type tumors (median: 0.99 vs. 0.34; *p* < 0.001). ctKRAS G12D mutation in ascites, miR-31-5, CA19-9, and CEA were not significantly different compared with wild type tumors (median: 0.39 vs. 0.41; *p* = 0.621, 0.18 vs. 0.08; *p* = 0.283, 4.72 vs. 3.10; *p* = 0.232, and 2.97 vs. 2.52; *p* = 0.443) (Figure 4). In this aspect, the tissue KRAS status affects the ctDNA fragments of buffy coat released into the bloodstream. There were notable differences in ctDNA KRAS levels depending on the tissue KRAS status. However, the mutations of NRAS and EGFR in tissues were not strongly linked with buffy coat ctKRAS G12D mutation, ascites ctKRAS G12D mutation, miR-31-5, CEA, and CA19-9, as shown in Appendix A. Our findings suggest that screening of ctDNA and miRNA will be a predictor for decision making in the early identification of patients, regarding who will have primary tumor DNA, and accordingly, who may benefit from adjuvant chemotherapies, such as cetuximab or bevacizumab for KRAS-mutated CRC patients. Patients with CRCs that had an unmethylation of MLH1 and MSH2 showed significantly higher buffy coat ctKRAS G12D mutation, ascites ctKRAS G12D mutation, miR-31-5, and mixed scores than the patients with methylation of MLH1 and MSH2. The patients with tissue MLH1 methylation (n = 64) had median values of 0.35, 0.26, 0.08, and −0.82, compared to 0.62, 0.76, 0.20, and −0.43 for the patients with tissue MLH1 unmethylation (n = 27), respectively (Figure 5).

The unmethylation vs. methylation of MLH1 was distinguished from those with buffy coat ctKRAS G12D mutation, ascites ctKRAS G12D mutation, miR-31-5, and mixed score (*p* = 0.006, *p* = 0.010, *p* = 0.001, and *p* = 0.001, respectively). Interestingly, the unmethylation status of MSH2 in tissue was also closely associated with buffy coat ctKRAS G12D mutation, ascites ctKRAS G12D mutation, miR-31-5, and mixed score. The patients with tissue MSH2 methylation (n = 65) had a median value of 0.36, 0.26, 0.08, and −0.81, compared to 0.62, 0.69, 0.22, and −0.43 for the patients with tissue MSH2 unmethylation (n = 26). The unmethylation vs. methylation of MSH2 was distinguished from those with buffy coat ctKRAS G12D mutation, ascites ctKRAS G12D mutation, miR-31-5, and mixed score (*p* = 0.013, *p* = 0.016, *p* < 0.001, and *p* = 0.001, respectively). However, CA19-9 and CEA in blood were not closely associated with the unmethylation vs. methylation of MLH1 and MSH2 in tissue. The median values of CA19-9 for patients in the methylation vs. unmethylation of MLH1 and MSH2 were 3.01 vs. 3.68 and 3.06 vs. 3.41. The median values of CEA for the patients in methylation and unmethylation of MLH1 and MSH2 were 2.85 vs. 2.09 and 2.88 vs. 2.01. A potential limitation of current study is the relatively small sample size of CRC patients. In the future, a larger sample size could be recruited for CRC patients, and including a healthy group, and patients with benign tumors would likely obtain more conclusive information and provide a better understanding for the stage-dependent correlation with ctDNA detection. Furthermore, clinical data from TNM stage, tumor differentiation, and survival ratio were not obtained, because of limited clinical information.

## 4. Conclusions

In this study, we proposed the specific gravity-free method, using a two-alginate-bead system, in order to isolate the ctDNA and exosome for predicting a primary tumor. Given their capturing properties, combining these two beads using approaches might provide a new method for a cancer diagnosis. Our study highlights the suitability of liquid biopsy as a non-invasive method for the prediction of CRCs and suggests a method that reflects an increased relative level of tumor DNA in the blood of patients with cancer. Our data suggest that ctDNA in the buffy coat can be used to identify the ctKRAS G12D mutation of CRCs for prediction of a primary tumor. These results indicate that cfDNA can provide comprehensive information on the biological status of the tumor burden.

## Figures and Tables

**Figure 1 micromachines-12-00987-f001:**
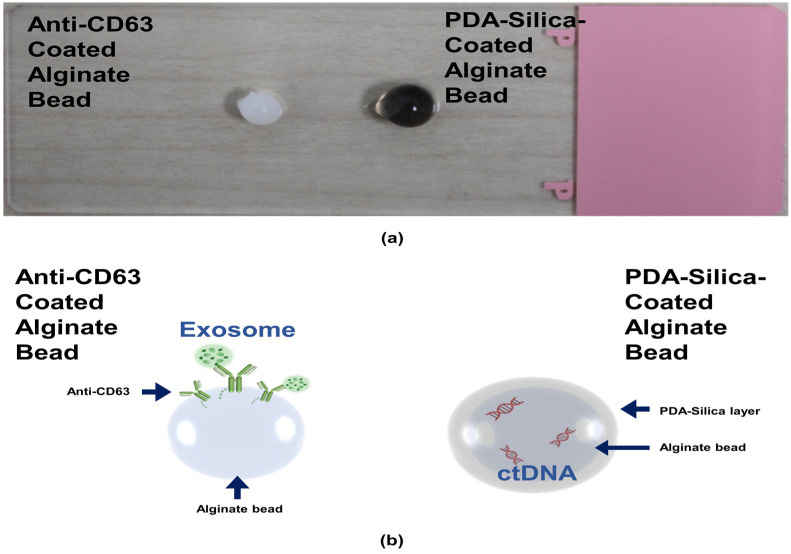
ctDNA isolated by a PDA–silica-coated alginate bead and an anti-CD63 immobilized alginate bead. (**a**) PDA–silica-coated alginate bead and anti-CD63 immobilized alginate bead, having a diameter of ~4 mm. (**b**) A schematic diagram illustrating applications of our isolation platform for ctDNA and exosome. It was used as a reliable platform for the diagnosis of CRC.

**Figure 2 micromachines-12-00987-f002:**
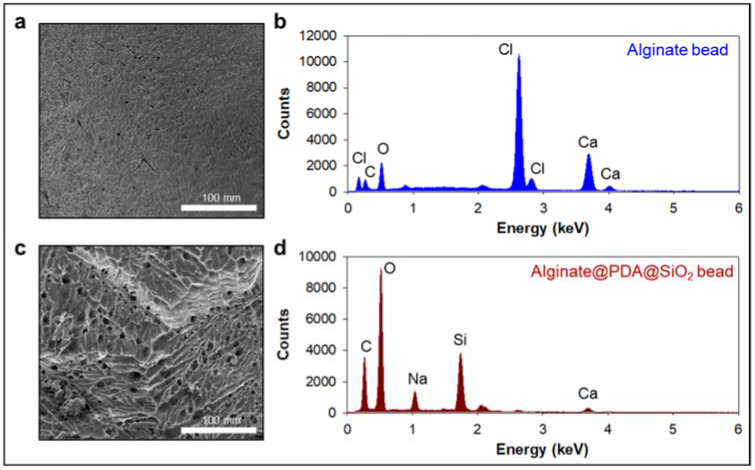
PDA–silica-coated alginate bead for enhanced ctDNA adsorption. (**a**,**c**) Morphological changes were confirmed by scanning electron microscopy (SEM) before (**a**) and after (**c**) PDA–silica-coated on alginate beads. The strong adhesion property of PDA enriched silica on the alginate surface before (**b**) and after (**d**) by EDS.

**Figure 3 micromachines-12-00987-f003:**
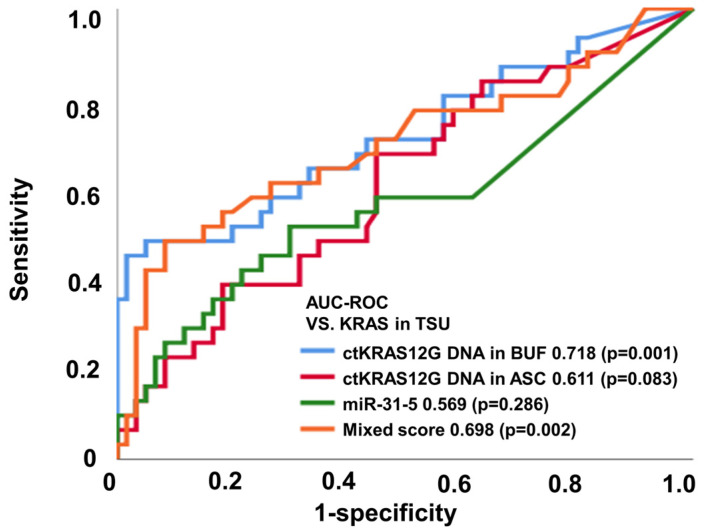
ctKRAS G12D mutation in the buffy coat, ctKRAS G12D mutation in ascites, miR-31-5 in exosome, and mixed score reflect the pathological status of the tumor burden based on KRAS.

**Figure 4 micromachines-12-00987-f004:**
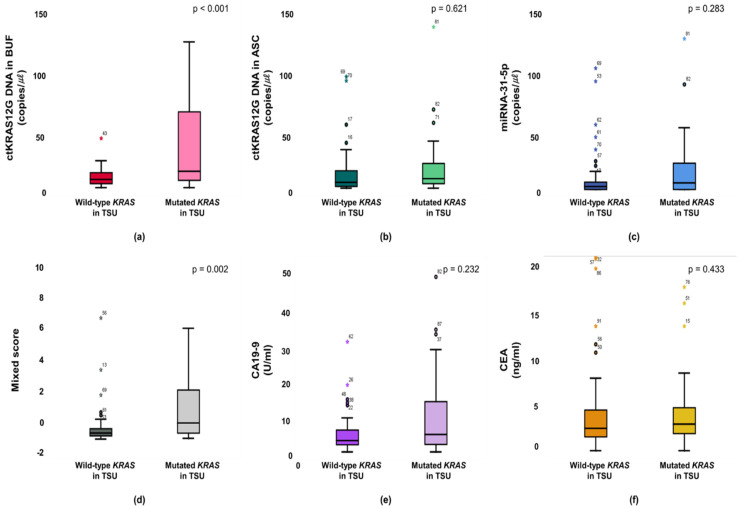
Primary statuses between mutant type and wild type KRAS are associated with the buffy coat (**a**), ascites (**b**), miR-31-5 (**c**), mixed score (**d**), CEA (**e**), and CA19-9 (**f**).

**Figure 5 micromachines-12-00987-f005:**
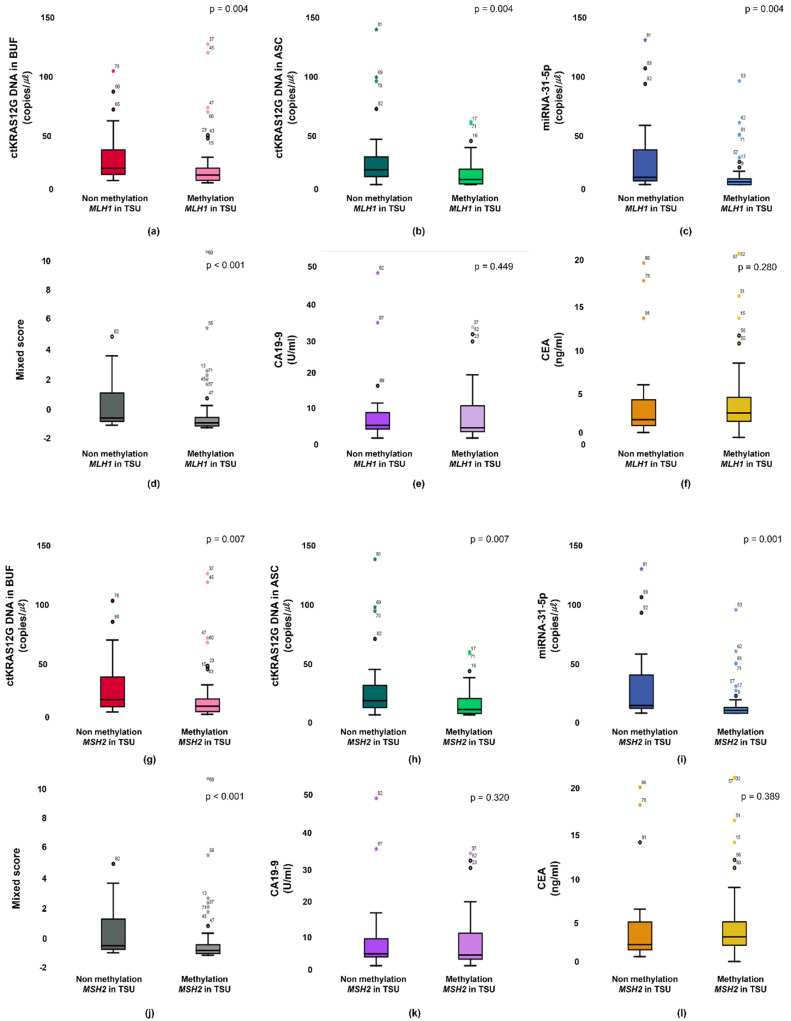
Primary status between methylation and non-methylation of MLH1 are associated from the buffy coat (**a**), ascites (**b**), miR-31-5 (**c**), mixed score (**d**), CEA (**e**), and CA19-9 (**f**). Primary status between methylation and non-methylation of MSH2 are associated from the buffy coat (**g**), ascites (**h**), miR-31-5 (**i**), mixed score (**j**), CEA (**k**), and CA19-9 (**l**).

**Table 1 micromachines-12-00987-t001:** Diagnostic capability of ctKRAS G12D and miR-31-5 analysis compared to KRAS, BRAF, MLH1, and MSH2 in tissue.

**KRAS mutant type in tissue** **vs. KRAS wild type in tissue**			**BRAF mutant type in tissue** **vs. BRAF wild type in tissue**		
**Variables**	**Pairwise** **comparison** **of AUROC** **(95% CI)**	***p* value**	**Variables**	**Pairwise** **comparison** **of AUROC** **(95% CI)**	***p* value**
ctKRAS DNA mutant in buffy coat	0.718 (0.598-0.838)	0.001	ctKRAS DNA mutant in buffy coat	0.540 (0.376-0.704)	0.786
ctKRAS DNA mutant in ascite	0.611 (0.489-0.734)	0.083	ctKRAS DNA mutant in ascite	0.193 (0.048-0.337)	0.038
miR-31-5 in exosome	0.569 (0.434-0.703)	0.286	miR-31-5 in exosome	0.293 (0.170-0.415)	0.002
Mixed score	0.698 (0.575-0.822)	0.002	Mixed score	0.269 (0.165-0.374)	0.001
**MLH1 methylation in tissue** **vs. no MLH1 methylation in tissue**			**MSH2 methylation in tissue** **vs. no MSH2 methylation in tissue**		
**Variables**	**Pairwise** **comparison** **of AUROC** **(95% CI)**	***p* value**	**Variables**	**Pairwise** **comparison** **of AUROC** **(95% CI)**	***p* value**
ctKRAS DNA mutant in buffy coat	0.300 (0.186-0.414)	0.003	ctKRAS DNA mutant in buffy coat	0.314 (0.197-0.430)	0.006
ctKRAS DNA mutant in ascite	0.277 (0.168-0.387)	0.001	ctKRAS DNA mutant in ascite	0.286 (0.174-0.398)	0.002
miR-31-5 in exosome	0.293 (0.170-0.415)	0.002	miR-31-5 in exosome	0.272 (0.152-0.392)	0.001
Mixed score	0.269 (0.165-0.374)	0.001	Mixed score	0.275 (0.169-0.381)	0.001

## Data Availability

All data generated from this study are included in this published article and Appendix A. Raw data are available from the corresponding author on reasonable request.

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
