# Peer review of "The Specific Gravity-Free Method for the Isolation of Circulating Tumor KRAS Mutant DNA and Exosome in Colorectal Cancer"

_micromachines, 2021, doi:10.3390/mi12080987_

Round 1

Reviewer 1 Report

In this manuscript, the authors present a dual-isolation methods that utilized two types of coated alginate beads to capture ctDNA and exosome from patient samples for early stage tumor diagnosis. The application and findings are novel, and this work is performed with decent and appropriate manner, however, the writings can be improved. The manuscript could be suitable for publication in Micromachines after addressing some minor concerns below.

Minors:

  1. Line 35, “so they have been might be frequently performed”, maybe “so they might have been frequently performed”?
  2. Line 79, “were previously coated”, maybe “were firstly coated”?
  3. Section 2.1, Please use a same acronym over the manuscript. D.W. or DW, please be consistent. (Line 83, Line 128)
  4. Line 100, Si3N4, need to do subscript, Si3N4
  5. Line 112, “it’s sample additiion to 200 ul…”, there is a typo, and also might be “it’s sample was added with 200 ul …”?
  6. Line 114, “Rnase/Dnase” should be “RNase/DNase”?   
  7. Number and unit should have a space in between. Line 113, Line 134
  8. Line 125, “in sample for remove phenol”, should be “in sample to remove phenol”?
  9. Line 126, “The sample was by …”, should be “The sample was further processed by …”?
  10. Line 163, Line 164, please use the same abbreviation of anti-CD63, or ab-CD63.
  11. Line 183, “the present study”, should be “presented?”
  12. Line 187, Line 188. 41.4% might be a wrong calculation, should it be 44/91 = 48.4%?
  13. Line 257, “will be predict for a decision making”, might be “will be a predict for decision making”?
  14. Line 258, “…patients who will be at primary tumor DNA”, should be “…patients regarding who will be with primary tumor DNA”?
  15. Line 263, “were median value of”, should be “were with median value of”?
  16. Figure 1, Ab-EPCAM, maybe was labeled wrong? Also, might be better to provide arrow to indicate each layer/drawing is corresponding to what chemical? Figure 1 caption (Line 157), “it use as …”, should be “it was used as …”?
  17. Figure 3, please put all your subtitles at the same location of each graph. Also, figure legends are shadowing some data presentations. Please reorganize the layout.
  18. Overall, please pay more attention to SI unit rules and styles conventions. You can refer to the following link for details: https://physics.nist.gov/cuu/Units/checklist.html

Specifically, unit “µl”, “µL” and “µl” are all presented in the same manuscript, that include section 2.3, section 2.4, Figure 4, Figure 5. Please be consistent with the format you use.

Author Response

Please see the attachment
We greatly appreciate the reviewer for such a favorable comment on our manuscript

Reviewer 2 Report

The authors describe a liquid biopsy for a specific KRAS mutation in colorectal cancer patients, and show that detection of that mutated ctDNA by ddPCR in peripheral blood correlates with the presence of the mutation in the solid tumor. The authors test for, but find only weak association between other liquid biopsies (KRAS DNA mutant in ascite tested by ddPCR, and miR-31-5 isolated using a separate bead designed to target exosomes, CA19-9 protein assay, and CEA protein assay) and several characteristics of the primary tumor (KRAS DNA mutant, MLH1 gene methylation, and MSH2 methylation). This reviewer has four major issues with the manuscript.

The largest issue with this manuscript is the authors’ interpretation of receiver operating characteristic (ROC) statistics. Every curve but two (ctKRAS12G DNA in BUF in Figure 3a and possibly “mixed score” in Figure 3a) show that the liquid biopsy is not predictive. In other words, an area under ROC (AUROC) of less than 0.5 is no better than random chance at prediction. The authors should consider any test with a confidence interval below 0.5 to be non-predictive. It is unclear how the authors generated a p value for these data or what the p value is comparing. In any case, it is inappropriate for this test. The authors should remove p values and consider all tests other than “ctKRAS G12D from buffy coat” testing for “KRAS G12D in tumor” to be non-predictive.

It is unclear what the authors mean by “Mixed score.” This should be defined in the methods.

The correlation analysis described in Figure S2 is flawed. Pearson’s correlation is an inappropriate test because the data are not normally distributed. The data should either be log-transformed and tested again for normality prior to running Pearson’s correlation, or nonparametric correlation should be used.

Certain important details for methods are missing. The authors should include methods for quantifying CA19-9 and CEA in the main text. The main text and figure S3 captions refer to mutant vs. wildtype EGFR, but Figure 3G-L appears to show expression level of EGFR from tumor immunohistochemistry. The authors should clarify the methods used to collect NRAS and EGFR expression data.

Author Response

(The authors gave the same response as above.)

Round 2

Reviewer 2 Report

Thank you for your edits. This reviewer's concerns were addressed in the updated version of the manuscript and in the authors' response.